# Exercise Dose Equalization in High-Intensity Interval Training: A Scoping Review

**DOI:** 10.3390/ijerph19094980

**Published:** 2022-04-20

**Authors:** Tom Normand-Gravier, Florian Britto, Thierry Launay, Andrew Renfree, Jean-François Toussaint, François-Denis Desgorces

**Affiliations:** 1Université Paris Cité, 75015 Paris, France; tom.normand-gravier@etu.u-paris.fr (T.N.-G.); florian.britto@u-paris.fr (F.B.); thierry.launay@u-paris.fr (T.L.); jean-francois.toussaint@aphp.fr (J.-F.T.); 2URP 7329-IRMES (Institute for Research in Medicine and Epidemiology of Sport), INSEP, 75012 Paris, France; 3Institute Cochin, U1016 INSERM, 75014 Paris, France; 4School of Sport & Exercise Science, University of Worcester, Worcester WR2 6AJ, UK; a.renfree@worc.ac.uk; 5CIMS, Hôtel-Dieu, Assistance Publique–Hôpitaux de Paris, 75004 Paris, France

**Keywords:** training programs, physical activity, effort, patients, athletes

## Abstract

Based on comparisons to moderate continuous exercise (MICT), high-intensity interval training (HIIT) is becoming a worldwide trend in physical exercise. This raises methodological questions related to equalization of exercise dose when comparing protocols. The present scoping review aims to identify in the literature the evidence for protocol equalization and the soundness of methods used for it. PubMed and Scopus databases were searched for original investigations comparing the effects of HIIT to MICT. A total of 2041 articles were identified, and 169 were included. Of these, 98 articles equalized protocols by utilizing energy-based methods or exercise volume (58 and 31 articles, respectively). No clear consensus for protocol equalization appears to have evolved over recent years. Prominent equalization methods consider the exercise dose (i.e., energy expenditure/production or total volume) in absolute values without considering the nonlinear nature of its relationship with duration. Exercises resulting from these methods induced maximal exertion in HIIT but low exertion in MICT. A key question is, therefore, whether exercise doses are best considered in absolute terms or relative to individual exercise maximums. If protocol equalization is accepted as an essential methodological prerequisite, it is hypothesized that comparison of program effects would be more accurate if exercise was quantified relative to intensity-related maximums.

## 1. Introduction

Exercise is both described and prescribed on the basis of two main variables: intensity (i.e., level of muscular activity) and volume (e.g., duration, distance or number of repetitions of an interval or set, and of the entire session) [1,2]. Notably for the interval exercise modality, these major variables also depend on possible recovery pauses within the exercise bout, inducing a third exercise variable, called by some authors “exercise density” (i.e., work/recovery ratio but also intensity level of the recovery) [1,3,4]. For quantifying and designating the overall exercise performed, authors can use generic terms accounting for all exercise variables, such as exercise dose in exercise-induced health studies [1] or training load for athlete monitoring purposes [5,6,7]. Defining effort as what is required to achieve a task in line with individual maximal capacities [8], exercise dose and training load might refer to the quantity of exercise-induced effort [5,6].

The control and calibration of training protocols should be a prerequisite in exercise and sport science studies, and insufficient consideration of this may result in confusion regarding exercise program effects [9,10]. Manipulation of training variables (volume, intensity and density) might ensure that the effort level generated by two protocols being compared is similar, or in other terms, that their exercise dose is equalized. However, methodologically these comparisons are not easy to conduct. Targeting large populations, recommendations for physical activity frequently use absolute values for intensity or, sometimes, exercise durations characterized by large intensity ranges (e.g., light, moderate, vigorous) that could complicate the quantification of an individualized and unique dose value [1]. Viana et al. suggested that conclusions about high-intensity interval training (HIIT) remain difficult to draw because of insufficient control of the numerous exercise variables [11]. Recently, the lack of protocol equalization in HIIT and moderate-intensity continuous training (MICT) has been suggested to represent a possible methodological bias limiting studies’ conclusions [12]. Comments on this paper suggest that consensus was not reached in the methods used for equalizing protocols nor, more surprisingly, in the necessity for equalizing them [13]. Limits and issues raised by equalization methods based on energy expenditure, although largely recommended, have only been recently reported [14,15]. Similar debates on adequate terms to use and on quantification methods are currently in progress regarding the training-load concept [2,7,16]. Therefore, we suggest that equalization of training doses should be a methodological prerequisite before comparing the effects of different training protocols and is therefore a major challenge facing exercise physiologists and sport scientists.

HIIT may be defined as repeated short-to-long exercise bouts performed at an intensity between 80% and 120% of maximum aerobic power (oxygen consumption or equivalent heart rate) [11]. Recently, the use of HIIT has been proposed as a method for improving quality of life of older people and for rehabilitation of patients suffering from several pathologies, such as cardiovascular diseases. As HIIT has become a real worldwide trend for exercise practice and sport sciences, this has increased the need for accurate equalization of training protocol doses in order to compare their efficiency [11,17,18]. Furthermore, we propose that HIIT studies display most of the characteristics necessary to understand the issues of exercise dose quantification and protocol equalization: (i) high number of studies published; (ii) changes in exercise variables; (iii) methods for equalization already developed and discussed.

The present scoping review aims to identify in the literature the evidence for protocol equalization and the soundness of methods used for it [19].

## 2. Materials and Methods

The latest methodological guidance for scoping reviews was followed, leading to completing the checklist of the Preferred Reporting Items for Systematic Reviews for scoping reviews (Appendix A) [20,21,22].

### 2.1. Search Strategy

We analyzed published studies on electronic databases until 30 November 2020 without restriction set on the publication year. PubMed and SCOPUS databases were explored using a keyword search strategy for ‘High-intensity interval training’ with a first filter step used for including studies that were: written in English; randomized controlled trials, clinical trials or from journal articles; based on human subjects. A second step was based on abstract screening to select studies comparing HIIT to another type of training program and to retain only chronic training programs. When the information was missing in the abstract, the authors searched for it in the whole article. Because variables measured to control exercise do not correspond between sprint interval training and HIIT, the last step consisted of retaining studies focusing on HIIT (80–120% of VO_2_max or equivalent) and excluding sprint interval training (intensity higher than 120%) [11]. All duplicate studies and protocols were excluded; if the same experimental protocol was used for several articles, only the first published was retained. Finally, studies were sorted according to publication year and type of subjects observed: (i) patients or older people; (ii) untrained; (iii) trained. All search results were extracted and imported into a reference manager (Zotero, version 5.0.96.3). No included studies were authored by any of the review authors, thereby limiting possible conflicts of interest.

### 2.2. Assessment of Reporting Quality

The reporting quality of studies was assessed using items specific to the research field. Most of them originated from a modified version of the Downs and Black checklist, resulting in eight assessment criteria (Appendix A) [23]. Studies reporting quality were scored on a scale from ‘0’ (unable to determine, or no) to ‘1’ (yes) for each item. Scores were allocated on the basis of good (6–8), moderate (3–5) and poor (0–2) methodological reporting quality.

### 2.3. Terms Used and Methods Applied for Protocols Equalization

Articles’ methods sections were analyzed, and the proportion of studies that equalized doses of training protocols and methods used for equalizing were recorded. If any information necessary for protocol equalization was not included in a study’s methods section, they were considered as not equalized. In line with this search, articles were analyzed to determine terms used to describe how exercise-induced effort was quantified (e.g., exercise dose or exercise volume) and the equalization process (e.g., equated protocols or matched training).

To assess the soundness of the methods used for protocol equalization, the exercise details were extracted from the articles, specifically exercise volume (duration, distance or number of repetitions for session and for each interval or set), intensity (varied metrics in absolute or relative values), recovery (duration and intensity if necessary) and exercise type (running, walking, cycling or resistance training).

### 2.4. Statistical Analysis

The present study is largely descriptive, and quantifies proportions (%) of studies that equalized training protocols and identified methods used for equalization. Differences in reporting-quality methodology between studies that equalized protocols and those that did not, and between subject populations were assessed by using one-way analysis of variance for total score and Pearson’s Chi-2 test for each criterion assessed. The evolution of dose equalization over the years was observed by linear regression analysis of percentage of studies equalizing doses. Statistical analysis was performed with R software (version 3.6.2), and statistical significance was set at *p* < 0.05.

## 3. Results

The identification process described in Figure 1 resulted in 169 studies being included in the review. The complete list of articles retained is presented in the Appendix A.

We aimed first to document the equalization of exercise protocols in studies comparing HIIT and other exercise types. We also aimed to highlight if protocol equalization was associated with a better-quality study design and/or if it was specific to recent studies.

The assessment of methodological reporting quality of these articles was moderate, but with poor quality for calculations of statistical power, and moderate for group homogeneity and for groups matched by physical condition (Table 1). Matching by “subjects’ physical condition” was the only criteria that led to a significant difference between types of subjects observed by studies (*p* < 0.001)

The most-frequently occurring terms used for the process of protocol equalization (total *n* = 98) were as follows: matched protocols (*n* = 44); equalized (or equated, equal, equivalent, *n* = 10); isocaloric (or isoenergetic, *n* = 8); The most-frequently used terms to designate what had been equalized were: total work (or external, mechanical, *n* = 26); workload (or training load, *n* = 29); exercise volume (or total volume, *n* = 13); exercise dose (or effort, *n* = 4). Protocol equalization did not evolve clearly over time, but there was a trend for a reduction in the proportion of equalizing studies (R^2^ = 0.21, *p* = 0.06; Figure 2) and an increase in the absolute number of studies that equalized protocols (R^2^ = 0.59, *p* = 0.01). No differences were observed in studies’ reporting quality between those that equalized protocol doses and those that did not (*p* = 0.1).

The distribution of studies based on equalized and non-equalized protocols, and associated methods for quantifying exercise doses are shown in Figure 3. Studies observing patients and older people equalized protocols at 58.7%, compared to 62.5% in untrained subjects and 51.7% in trained, without significant differences between groups (*p* = 0.09). Training protocols differed between studies; however, typical HIIT exercises were identified among all studies (Figure 3).

## 4. Discussion

We aimed to determine whether researchers, when comparing HIIT to other types of programs, had utilized equalized protocols. Although most studies equalized protocols, a substantial number of studies did not. For designating what was equalized, authors mainly focused on actual measures performed (e.g., total work, energy expenditure or exercise volume) rather than using a more generic term (e.g., exercise dose, training load or effort). Energy-based methods were prominently used for equalizing protocols, whereas methods based on exercise volume and perceived exertion appeared markedly less frequently.

Among the 169 studies included in this review paper, most equalized their protocols (58%), whilst 42% did not. Consensus for protocol equalization is not apparent, and the protocol equalization rate has not evolved significantly since the first paper published in 1979. In addition, data did not show differences according to populations observed. This is in line with the assessment of reporting quality, which did not differentiate studies according to protocol equalization or populations observed. Satisfactorily, “exercise control” and “direct supervision” criteria of reporting quality achieved the highest assessments. Among studies that did not equalize protocols, twenty-one compared HIIT with typical MICT programs (Figure 3) that had been designated by previous studies to be equal based on energy expenditure or production [24]. Therefore, although protocol equalization was not reported in the methods section of these studies, it had possibly been achieved anyway, thereby increasing the proportion of protocols actually equalized.

Vollard and Metcalf [13] argued that the key advantage of HIIT is time efficiency. MICT requires more prolonged exercise duration than HIIT, and it could be presumed as self-evident that MICT is not as effective if exercise duration is short. However, for a given exercise duration, because of higher intensity, HIIT induces a greater exercise dose than MICT. If the aim is to demonstrate the positive effects of HIIT despite a short exercise duration, such demonstration could be achieved without requiring comparison with another training program. Conversely, when comparing programs’ effects on performance improvement or biological parameters, if the higher exercise intensity of HIIT is not counterbalanced by a lower exercise volume, responses may have originated from the higher intensity, but also simply from a greater exercise dose. This methodological point was accounted for by 98 studies that attempted to equalize protocols.

In some studies, training protocols were partly equalized by prescribing similar total exercise durations. Such a method is in line with population-based studies that quantify physical activity through time spent in light/moderate/strenuous intensity ranges without aiming to compare the particular effects of these intensity levels [1]. Using session durations to equalize protocols corresponded to physical activity recommendations for health and wellbeing (e.g., three sessions of 30–45 min per week for HIIT and moderate intensities) [1]. During HIIT, high-intensity activity itself could not account for the entire 30–45 min of the session: 10–20 min of high-intensity exercise was paired with low-intensity exercise for the remaining 10–20 min. Therefore, protocols equalized by similar durations compared MICT to mixed MICT and HIIT, but studies did not describe the rationale underpinning the selection of exercise durations for different intensities. Equalization by total volume does not consider the slope of the relationship between intensity and duration, and even less the nonlinearity of this relationship. Consequently, the absolute value of exercise duration was equalized, but not the combination of the exercise variables. If expressed relative to respective maximums, durations prescribed by HIIT programs were markedly higher than for MICT. In these studies, responses to training might be due to changes in intensity or to changes in exercise dose. Furthermore, by proposing similar exercise durations, these protocols cancelled the time gains expected from HIIT [13].

The primary methods used for protocol equalization were energy-based. Most studies measured exercise-induced energy expenditure through oxygen consumption, while some others measured external work based on power output and exercise duration [12,24,25]. Energy expenditure methods typically incorporated both exercise and recovery periods, while methods based on external work only considered exercise bouts. That is quite surprising as the typical HIIT exercise utilized in studies based on external work (i.e., 8–10 × 1 min at 90–95% HRmax, 1–2 min recovery) were characterized by short–moderate recovery pauses, allowing maintenance of a high level of physiological stress [26]. Furthermore, exercise-induced excess post-oxygen consumption is largely influenced by exercise intensity and may be prolonged for many hours [27]. Although some authors suggest that exercise-induced energy expenditure should also account for exercise-induced excess post-oxygen consumption, this point may require more careful attention in HIIT studies that focus on the effects of changes in both intensity and interval volumes [1,12].

Energy-based methods for quantifying exercise consider the human ability for energy expenditure or external work to be similar whatever the exercise intensity. For several decades, models of the intensity–volume relationship have described a hyperbolic pattern, with maximal exercise volumes dramatically decreasing with increases in intensity [28,29,30]. By extension, maximal energy expenditure/external work follows the same pattern [31]. Thanks to recovery pauses, for a given intensity level, interval exercise allows accumulation of more exercise than continuous exercise and, consequently, greater energy expenditure [26]. The typical 4 × 4 min session is likely to be performed at a higher intensity level than a 16 min exercise performed in continuous modality [30]. Seiler et al. reported that the maximal tolerable intensity for 4 × 4 min was 94 ± 2% of maximal heart rate when interspersed with 2 min passive recovery [32]; in HIIT studies, an active recovery (3 min at 70% HRmax) was added to this maximal effort. Conversely, because of the nonlinear relationship between exercise intensity and energy expenditure, typical MICT exercise appears to be far from the exercise dose performed during typical HIIT. In fact, in typical MICT, 30–45 min is prescribed at 65–75% HRmax, an intensity that can be maintained for several hours before exhaustion. It may be assumed that the typical HIIT exercise resulting from the energy-based equalizing method reached a maximum of energy expenditure and was exhausting, while MICT represented relatively easy training. This assumption is supported by significantly higher ratings of perceived exertion (RPE) following HIIT sessions [25,33,34], and some authors argued that energy-based methods for equalization underestimate the work that athletes are able to perform at lower intensities [32,35]. Such differences in session-induced exertion should be considered as a possible methodological bias that is likely to become more pronounced with increases in intensity differences between programs. HIIT-induced dose could represent the maximum tolerable (or excessive) training stimulus, whereas MICT dose could be low or insufficient. Finally, despite the popularity of equalization methods based on energy expenditure, its soundness and relevancy are still questioned [13,15].

Finally, six studies used RPE to equalize protocols, and only one used the session-RPE-based method for training-load quantification (i.e., duration × RPE of the session). It seems that studies equalizing protocols by using RPE were composed of varied exercise modalities (e.g., running, resistance exercise or skating) [36,37,38]. RPE is not only influenced by exercise intensity [39], as exercise duration [40,41], interval volume [42], exercise modality [43] and recovery periods [44] have also been reported to significantly influence RPE. Finally, RPE appears to be influenced by all exercise variables and, consequently, might represent a subjective assessment of the exercise dose. Previous studies have shown that it provides similar session assessments to exercise volume expressed relative to maximum for the considered intensity level [4,40]. Conversely, training load based on RPE might account twice for the exercise volume (i.e., in duration and RPE itself), inflating the calculated load for prolonged sessions [4,5]. In line with studies that have used RPE for protocol equalization, some authors have suggested that RPE alone is therefore preferable for exercise quantification, thereby avoiding the overexpression of volume [4,45].

We acknowledge that the present study may have overlooked some published papers, as it was only conducted on two literature databases and only considered original experimental investigations. Based on the numerous studies utilizing equalization of protocols and researcher support for equalization, it seems that, although the need for equalization is not debated per se, the soundness of methods for equalizing is [1,5,7,12]. In addition, generic terms that designate the quantity of exercise-induced effort (i.e., exercise dose, internal training load) and associated quantification methods (e.g., RPE) may be considered to account for individual maximal capacities in the exercise considered [5,6,7,8]. In essence, this is not the case for energy expenditure/production or exercise volume. Finally, the main methodological issue is whether to quantify the exercise—whatever the method—in absolute values or relative to individual maximums for the considered exercise. Lack of consideration of the slope and nonlinearity of the energy–duration or of the intensity–duration relationship is questionable. As proposed recently for training-load quantification and by one study among the 169 retained [4,46], we hypothesize that exercise quantified relative to maximum energy expenditure/external work, or exercise volume, for specified intensity levels will allow more precise program comparisons. This may also be the case when dose is assessed via perceived exertion.

Although scoping reviews can be the first step before systematic review or meta-analysis on the topic, and even if only equalized protocols were retained, results of studies comparing HIIT vs. MICT should be interpreted carefully because of the uncertain accuracy of equalization methods mainly used [19].

## 5. Conclusions

In HIIT studies, no clear consensus for protocol equalization appears to exist, and there has been no evolution in practices over time. If the scientific community supports this methodological prerequisite, it may assist with the assessment of methodology reporting quality.

Equalization based on exercise duration does not consider all the variables composing exercise-induced effort. Primary equalization methods consider energy expenditure/external work in raw values without considering the slope and the nonlinear nature of its relationship with duration. Exercises resulting from these quantification methods induced maximal exertion in HIIT exercises but low exertion in MICT. Evidently, the main issue is whether to consider exercise dose in absolute values or relative to individual exercise maximums. It is hypothesized that comparison of program effects would be more accurate if the exercise (e.g., exercise volume, energy) was expressed relative to intensity-related maximums (e.g., perceived exertion, exercise volume relative to maximum).

## Figures and Tables

**Figure 1 ijerph-19-04980-f001:**
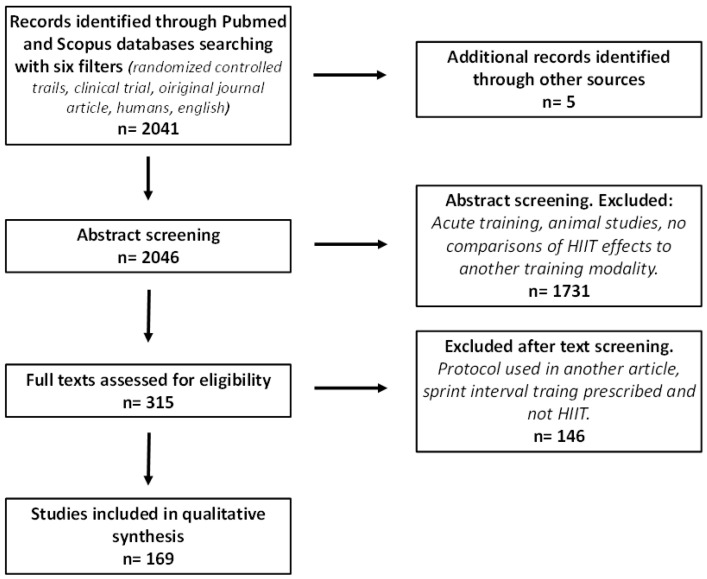
Phases of study selection during data collection.

**Figure 2 ijerph-19-04980-f002:**
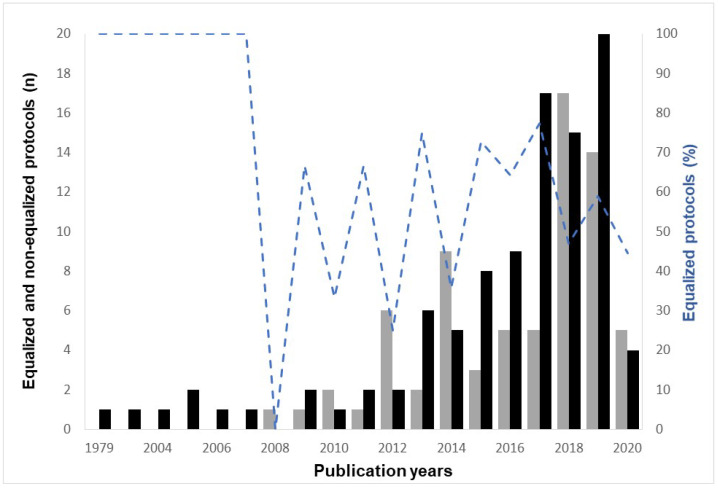
Percentage and absolute number of studies using equalized protocols (dashed blue line and black bars, respectively) and number of studies without using equalization of protocols (grey bars) from 1979 to November 2020.

**Figure 3 ijerph-19-04980-f003:**
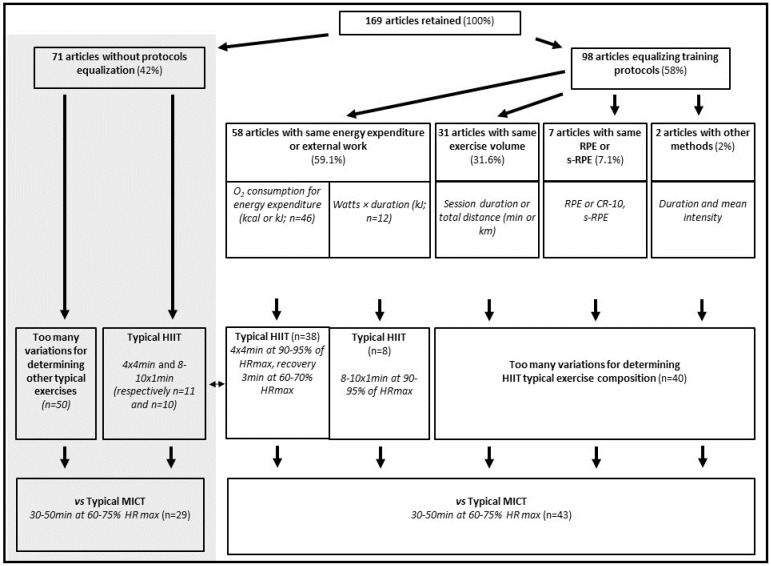
From article-selection process to equalization methods and exercise sessions.

**Table 1 ijerph-19-04980-t001:** Reporting quality expressed through positive assessment of studies according to protocol equalization processes (middle of table) and population observed (bottom of table). Total score expressed as mean and standard deviations.

	Total Score	Recruitment in Same Population (%)	Subjects Ramdomization (%)	Physical Condition Matching (%)	Training Direct Supervision (%)	Exercise Control (%)	Adherence to Training (%)	Subjects Follow-Up (%)	Statistical Power (%)
Total (*n* = 169)	5.1 ± 1.5	47.6	70.8	53.0	73.8	88.7	61.3	84.5	29.8
Equalized protocols (*n* = 98)	5.2 ± 1.5	46.4	71.1	51.5	75.3	91.7	61.9	89.7	28.9
Non-equalized protocols (*n* = 71)	5.0 ± 1.6	49.3	70.4	54.9	71.8	84.5	60.5	77.4	31.0
Older people and patients (*n* = 99)	5.1 ± 1.7	50.5	67.0	44.4	76.7	85.8	62.6	85.8	29.3
Untrained (*n* = 41)	5.3 ± 1.3	52.5	85.0	47.5	75.0	92.5	57.5	87.5	32.5
Trained (*n* = 29)	5.0 ± 1.2	34.5	62.1	89.6 *	62.1	93.1	62.1	75.9	24.1

* significant differences with other groups of subjects (*p* < 0.05).

## Data Availability

Not applicable.

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
