# Peer review of "Exercise Dose Equalization in High-Intensity Interval Training: A Scoping Review"

_ijerph, 2022, doi:10.3390/ijerph19094980_

Round 1

Reviewer 1 Report

General Comments:

Many thanks to the authors for addressing previous reviewer comments. The quality of the manuscript has been somewhat improved in this new version. However, there remains to be a lack of clarity and flow in the way the paper is written, and it can be quite difficult to read at times. The discussion section in particular is quite lengthy and could be written in a much more concise manner while still outlining the key points without repetition.

Specific Comments:

Lines 32-34: This sentence is unnecessary

Line 39: Should this external training load rather than internal, as it is referring to the external exercise stimulus imposed on an athlete rather than their individual responses to a given stimulus or dose.

Line 64: Who specifically is it a major challenge for? It seems that it would be a challenge for clinical exercise physiologists or researchers seeking to determine whether HIIT or MICT is most effective in improving various factors relating to cardiorespiratory fitness. It seems to be less of an issue for sport scientists working with elite populations, as they will typically complete both forms of training plus several others.

Line 70: Refer to point above, this is more of a clinical/exercise/health trend, as elite athletes/coaches have been performing HIIT for years, but not for the same outcomes as the old people or patients as you mention earlier in this paragraph.

Table 1: The formatting of this table still requires significant improvement, but perhaps it is the way it has been formatted in this proof. Inconsistent decimal points for total score.

Line 151: Remove ‘that’

Lines 151-152: What is the difference between total work or external/mechanical work, and workload or training load?

Line 154: Does R2 refer to R squared value?

Figure 2: The presentation of this figure still needs improving as per previous comments. Adding in axis lines would be helpful. It remains difficult to read/interpret.

Lines 168-169: Unsure why this text has come up significantly larger on this version, please amend.

Line 181: Unsure if there is a need for the sub-headings within the discussion.

Line 190: What was the active recovery period for the 4x4 min? Also, 30-50 min is quite a large range and would no doubt produce quite large differences in energy expenditure.

Lines 193-195: Are you suggesting here that protocols may not have explicitly equalized protocols, but in fact they may have been equalized but we have no way of knowing as they have not measured this or specified this in their methods section? If so, the clarity of this sentence could be improved and should be part of the previous paragraph as it is referring to the above point re. studies that did not technically equalize protocols.

Line 256: This is assuming that volume and duration are the same thing?

Line 272-273: Perhaps expand on this a little to briefly outline that recovery periods allow for a higher intensity to be achieved in the work period, enabling the accumulation of a greater time at higher intensities than MICT. This is related to the recovery of various physiological variables e.g., PCr resynthesis.

Lines 276-278: So this is 4x4 min max – 3 min recovery, totalling 28 min? Is the intensity prescription for these efforts maximal effort as indicated?

Line 279: Avoid using the term ‘overreaching’ here, as this refers to a completely different phenomenon.

Line 322: HIIT is referred to as maximal exertion here, and earlier on in the paper as well. But based on your definition of HIIT within the introduction, HIIT isn’t always necessarily maximal in exertion, but rather just high intensity, above a certain threshold.

Author Response

Authors responses appeared below the comments and in italics.

General Comments:

Many thanks to the authors for addressing previous reviewer comments. The quality of the manuscript has been somewhat improved in this new version. However, there remains to be a lack of clarity and flow in the way the paper is written, and it can be quite difficult to read at times. The discussion section in particular is quite lengthy and could be written in a much more concise manner while still outlining the key points without repetition.

            The text has been reviewed entirely, the discussion has been reorganized and a bit shorten.

Specific Comments:

Lines 32-34: This sentence is unnecessary

            Sentence removed.

Line 39: Should this external training load rather than internal, as it is referring to the external exercise stimulus imposed on an athlete rather than their individual responses to a given stimulus or dose.

            Effectively it may be confusing to add to training load the adjective " internal" or external. We choose to remove internal for a more simple and generic "training load". See response for Lines 151-152 comment.

Line 64: Who specifically is it a major challenge for? It seems that it would be a challenge for clinical exercise physiologists or researchers seeking to determine whether HIIT or MICT is most effective in improving various factors relating to cardiorespiratory fitness. It seems to be less of an issue for sport scientists working with elite populations, as they will typically complete both forms of training plus several others.

            This paragraph is not exclusively based on HIIT vs MICT and deals with exercise quantification.

            We agree that HIIT vs MICT suggest that training programs were performed exclusively in the expected intensity ranges (i.e. mainly for MICT as HIIT require warm-up and cool down at low-moderate intensities).

           Few protocols compared training programming effects in elite athletes' population but they could have induced some debates on training intensity distributions (TID). It can be assumed that effects of TID might be compared on the basis of equalized protocols doses.

Line 70: Refer to point above, this is more of a clinical/exercise/health trend, as elite athletes/coaches have been performing HIIT for years, but not for the same outcomes as the old people or patients as you mention earlier in this paragraph.

            We agree that HIIT vs MICT mainly concerns exercise for health purposes.  

Table 1: The formatting of this table still requires significant improvement, but perhaps it is the way it has been formatted in this proof. Inconsistent decimal points for total score.

            Format and column headings improved

Line 151: Remove ‘that’

            Removed.

Lines 151-152: What is the difference between total work or external/mechanical work, and workload or training load?

            We consider only terms and definitions provided by authors (very frequently terms were not defined): - total work mainly refers to mechanical measures in kJ; workload and training load mainly terms used in the training field, workload describing variables of exercise such as distance, durations, or running speeds considered separately; training load might refer to a combination of these variable influences into a single amount.

  Line 154: Does R2 refer to R squared value?

            R2 changed for R2

Figure 2: The presentation of this figure still needs improving as per previous comments. Adding in axis lines would be helpful. It remains difficult to read/interpret.

            Figure improved notably axis titles, evolution line and corresponding axis changed in blue colour.

Lines 168-169: Unsure why this text has come up significantly larger on this version, please amend.

            Provided by the editorial formatting as it was; change made.

Line 181: Unsure if there is a need for the sub-headings within the discussion.

            Subheading and discussion reformatted.

Line 190: What was the active recovery period for the 4x4 min? Also, 30-50 min is quite a large range and would no doubt produce quite large differences in energy expenditure.

   The sentence has been slightly changed by referring to figure 3. Recovery was performed at 70% of HRmax (fig 3). We choose to present high range of duration (and the corresponding high range of intensity, VO2 %) to not present different MICT according to energy expenditure and energy production to gather a significant number of articles and clarify figure 3.

Lines 193-195: Are you suggesting here that protocols may not have explicitly equalized protocols, but in fact they may have been equalized but we have no way of knowing as they have not measured this or specified this in their methods section? If so, the clarity of this sentence could be improved and should be part of the previous paragraph as it is referring to the above point re. studies that did not technically equalize protocols.

            Added to the previous paragraph and sentence slightly modified.

Line 256: This is assuming that volume and duration are the same thing?

      Effectively we assumed that duration is a way for measuring exercise quantity and this quantity can be called volume. We do not agree with the fact that volume is exclusively defined by duration×intensity (as stated by ACSM) which provides -in our opinion a total volume- but duration or distance of an interval is -in our opinion- the first measure of the volume (i.e. of the interval).

Line 272-273: Perhaps expand on this a little to briefly outline that recovery periods allow for a higher intensity to be achieved in the work period, enabling the accumulation of a greater time at higher intensities than MICT. This is related to the recovery of various physiological variables e.g., PCr resynthesis.

      We agree that recovery periods allow to partially recover (!), and so allow to enhance the accumulated exercise volume at the same intensity or to enhance the intensity. Sentences slightly modified (Lines 255-262 of modified version with apparent changes).

Lines 276-278: So this is 4x4 min max – 3 min recovery, totalling 28 min? Is the intensity prescription for these efforts maximal effort as indicated?

       Articles frequently stated that recovery pauses were between intervals, so certainly three recovery pauses and a total exercise duration of 25 min; few papers reported the actual intensity maintained or the number of intervals actually performed. However, as exercise intensity was mostly controlled by HR in these studies, it remains possible that speeds or power output have been reduced while HR was maintained in the expected range.

Line 279: Avoid using the term ‘overreaching’ here, as this refers to a completely different phenomenon.

            Sentences slightly modified.

Line 322: HIIT is referred to as maximal exertion here, and earlier on in the paper as well. But based on your definition of HIIT within the introduction, HIIT isn’t always necessarily maximal in exertion, but rather just high intensity, above a certain threshold

   We agree and support that HIIT has to be defined through intensity of exercise. What is stated in this paper is that HIIT protocols resulting from equalization methods were exhausting and largely differ to what was expected from MICT. The level of exertion achieved might be a way to assess the exercise dose and should not present such differences in protocols compared (exertion might not be assessed exclusively by intensity, but also by duration, density...).

Reviewer 2 Report

The manuscript has been greatly improved.

Minor revisions are required:

  1. Improvement in tables, graphics, and figures quality is needed.
  2. Researchgate is a platform to report manuscripts published by each member. If the protocol review was published in a scientific journal, the authors could report the adequate reference. If not, I suggest rewriting the statement without researchgate mention.
  3. Lines 139-142 must be formatted according to the author guidelines journal.

Author Response

Authors responses to comments below the comment and in italics.

The manuscript has been greatly improved.

Minor revisions are required:

  1. Improvement in tables, graphics, and figures quality is needed.

                        Figures and tables have been improved.

2. Researchgate is a platform to report manuscripts published by each member. If the protocol review was published in a scientific journal, the authors could report the adequate reference. If not, I suggest rewriting the statement without researchgate mention.

      Guidelines for scoping reviews suggested that protocols deposit might be indicated by studies. However, for a better clarity of the text we remove the mention to researchgate.

3.Lines 139-142 must be formatted according to the author guidelines journal.

       Italics removed.

Round 2

Reviewer 1 Report

Thanks for all of your responses. No further ammendments are advised.

This manuscript is a resubmission of an earlier submission. The following is a list of the peer review reports and author responses from that submission.

Round 1

Reviewer 1 Report

General Comments:

This review is quite informative and provides a brief overview of the literature regarding exercise dose equalization in HIIT. Extensive scoping review is included in the title of this paper; however, it lacks detail and objective outcomes. While there is a fairly good argument put forward to support the need for exercise dose equalization and this review, this could be presented more clearly by relating it more closely to the real-world applications of HIIT and how it would help practitioners and researchers in the future. Additionally, a clear and concise definition of what you mean by the term equalization early on would likely be helpful for readers with little background in this area. The authors make some suggestions regarding methods for equalization in the conclusion, however it would be useful to provide more details on how researchers should approach this in future studies based on the outcomes of this review.

Specific Comments:

Line 34: Alter ‘suffers’ to ‘suffer’

Line 38: Suggest change in wording to ‘quantify the exercise does precisely’

Line 40: Missing word between believe and essential e.g., ‘We believe it is essential to…’

Line 41: Missing word between accounts and possible e.g., ‘accounts for possible’

Lines 90-92: Could you please state why sprint interval training was excluded? It has been stated that it is at an intensity above 120% of VO2max, and it is my understanding that SIT forms a component of HIIT. If this is not the case, please provide a supporting reference.

Table 1: The presentation of this table could be much improved. I assume total score is mean plus or minus SD, but this isn’t specified in the table title. Also please consider standardising the decimal points e.g., 67 or 67.0

Figure 2: A clear key for this figure would be helpful for interpretation rather than needing to re-read the figure title numerous times. Additionally, the quality of the figure in terms of clarity of text in the version I can see needs improving prior to publication.

Figure 3: Similar to above, clarity of text could be improved prior to publication.

Line 164: Change ‘lesser’ to ‘less’

Lines 178-179: This last sentence in the paragraph is unclear. Please consider re-phrasing.

Lines 180-188: This paragraph is worded unclearly and is therefore may be confusing to some readers. I’d suggest re-wording this paragraph to highlight your key points more clearly. It doesn’t flow on well from the previous paragraph.

Lines 196-198: Avoid using the word obviously. Also, I understand what you mean by this sentence, however being a bit more explicit in terms of why HIIT would not last the entire 30-45 min and maybe how much of that would have been actual work time to highlight the difference.

Line 207: Please expand on why oxygen consumption might better reflect the energetic demands of HIIT given the recovery periods. While the reasons are implied, it could be better explained by expanding this sentence slightly.

Lines 210-211: Consider re-wording this sentence.

Lines 222-230: This paragraph is confusing and unclear. Please consider re-wording this to better explain the benefits and limitations of the RPE method.

Lines 244-245: Should this say accumulation of more exercise at a greater intensity, rather than just more exercise given the total duration would be less with HIIT than continuous exercise.

Author Response

Reviewer 1.

General Comments:

This review is quite informative and provides a brief overview of the literature regarding exercise dose equalization in HIIT. Extensive scoping review is included in the title of this paper; however, it lacks detail and objective outcomes. While there is a fairly good argument put forward to support the need for exercise dose equalization and this review, this could be presented more clearly by relating it more closely to the real-world applications of HIIT and how it would help practitioners and researchers in the future.

            A paragraph dealing with practical applications has been added at the end of the discussion,          we think it really improves the paper as suggested (lines 306-324).

Additionally, a clear and concise definition of what you mean by the term equalization early on would likely be helpful for readers with little background in this area.

            Added in the introduction (lines 46-48): "The equalization of exercise doses effectively refers to the process   of manipulating training variables (volume, intensity and density) in an attempt       to ensure that the stimulus generated by two protocols being compared is similar."

The authors make some suggestions regarding methods for equalization in the conclusion, however it would be useful to provide more details on how researchers should approach this in future studies based on the outcomes of this review.

            A paragraph dealing with practical applications has been added at the end of the discussion,          we think it really improves the paper as suggested (lines 306-324).

 Specific Comments:

Line 34: Alter ‘suffers’ to ‘suffer’

            Corrected

Line 38: Suggest change in wording to ‘quantify the exercise does precisely’

            Corrected

Line 40: Missing word between believe and essential e.g., ‘We believe it is essential to…’

            Added

Line 41: Missing word between accounts and possible e.g., ‘accounts for possible’

            Added

Lines 90-92: Could you please state why sprint interval training was excluded? It has been stated that it is at an intensity above 120% of VO2max, and it is my understanding that SIT forms a component of HIIT. If this is not the case, please provide a supporting reference.

            The reference used in the introduction was added for this sentence (Viana et al,      2018).   Several authors assume that HIIT and SIT differ and then SIT should not be considered as          part of HIIT; for this reason and variables measured to control exercise do not correspond     between sprint interval training and HIIT (i.e. SIT are frequently performed all-out and then      intensity of intervals is decreasing over the sessions) we excluded SIT studies. Added line 99-100.

Table 1: The presentation of this table could be much improved. I assume total score is mean plus or minus SD, but this isn’t specified in the table title. Also please consider standardising the decimal points e.g., 67 or 67.0

            Mean and SD added in the title. Corrected

Figure 2: A clear key for this figure would be helpful for interpretation rather than needing to re-read the figure title numerous times. Additionally, the quality of the figure in terms of clarity of text in the version I can see needs improving prior to publication.

            Legend added within the figure improved, and figure pasted with best quality and clarity.

Figure 3: Similar to above, clarity of text could be improved prior to publication.

            Figure pasted with best quality and clarity.

Line 164: Change ‘lesser’ to ‘less’

            Corrected, line 180

Lines 178-179: This last sentence in the paragraph is unclear. Please consider re-phrasing.

            End of the sentence modified line 194-196. " … it was possibly achieved anyway,          thereby increasing the proportion of protocols actually equalized."

Lines 180-188: This paragraph is worded unclearly and is therefore may be confusing to some readers. I’d suggest re-wording this paragraph to highlight your key points more clearly. It doesn’t flow on well from the previous paragraph.

            Paragraph re-worded, line 198-208.

Lines 196-198: Avoid using the word obviously. Also, I understand what you mean by this sentence, however being a bit more explicit in terms of why HIIT would not last the entire 30-45 min and maybe how much of that would have been actual work time to highlight the difference.

            Sentence improved, Lines 225-229

Line 207: Please expand on why oxygen consumption might better reflect the energetic demands of HIIT given the recovery periods. While the reasons are implied, it could be better explained by expanding this sentence slightly.

            Sentence slightly modified, lines 239-243

Lines 210-211: Consider re-wording this sentence.

            Sentence modified, lines 246-250.

Lines 222-230: This paragraph is confusing and unclear. Please consider re-wording this to better explain the benefits and limitations of the RPE method.

            Paragraph re-worded, lines 262-272.

Lines 244-245: Should this say accumulation of more exercise at a greater intensity, rather than just more exercise given the total duration would be less with HIIT than continuous exercise.

            Sentence slightly modified, lines 303-307.

Reviewer 2 Report

1. Authors must follow the PRISMA checklist/protocol carefully: http://prisma-statement.org/documents/PRISMA_2020_checklist.pdf

2. The main concern of this study is evaluating different methods for exercising dose equalization/quantification in literature without considering the results of the manuscripts. 

3. There is a lack of information about limitations and biases of the study. 

4. Improvement in tables, graphics, and figures quality is required.

Author Response

Reviewer 2

Comments and Suggestions for Authors

  1. Authors must follow the PRISMA checklist/protocol carefully: http://prisma-statement.org/documents/PRISMA_2020_checklist.pdf

"The PRISMA 2020 statement has been designed primarily for systematic reviews of studies that evaluate the effects of health interventions, irrespective of the design of the included studies." As added at the end of introduction we assume that if dose equalization is lacking or inaccurate, results and conclusions are biased. The first purpose of the present study is likely the contrary of PRISMA statement: not evaluate the effects of interventions but only methods used for the prerequisite of exercise dose equalization that leads to study designs. Consequently, PRISMA checklist cannot be applied entirely notably the part of checklist dedicated to results.

Presentation improved lines 86-90

  1. The main concern of this study is evaluating different methods for exercising dose equalization/quantification in literature without considering the results of the manuscripts. 

  Added at the end of introduction lines 76-79. We propose that a lack of efficient equalization methods for the exercise dose represents a major methodological bias in studies comparing HIIT to MICT, thereby limiting the soundness of their results and conclusions, and enhancing the need to focus on dose quantification issue.

  1. There is a lack of information about limitations and biases of the study. 

            A paragraph dealing with limits of the study (lines 306-309) and practical applications has been added at the end of the discussion (lines 310-324).

  1. Improvement in tables, graphics, and figures quality is required.

            Table 1 standardized, figure 1 legend improved and clarity of figures 2-3 enhanced as requested by reviewer 1.

Round 2

Reviewer 1 Report

Thank you to the authors for addressing these comments. While the changes made have slightly improved some of the clarity and language issues within the manuscript, there remains to be a lot of work needed before this is ready for publication. It is difficult to read and there are many limitations as you now acknowledge at the end of the discussion. Additionally, the relevance and contribution of this study to the current body of research in this field is quite low.

Reviewer 2 Report

Thank you to the authors for addressing those questions. The new version of the manuscript presents a slight quality improvement. However, the type of review remains confusing. Please see lines 86 - 89. Some scoping reviews require a priori protocol.

Systematic review or scoping review? Guidance for authors when choosing between a systematic or scoping review approach: https://doi.org/10.1186/s12874-018-0611-x

Characteristics of Indigenous primary health care models of service delivery: a scoping review protocol: doi: 10.11124/jbisrir-2015-2474